# ZmDRR206 Regulates Nutrient Accumulation in Endosperm through Its Role in Cell Wall Biogenesis during Maize Kernel Development

**DOI:** 10.3390/ijms24108735

**Published:** 2023-05-13

**Authors:** Yanmei Li, Dongdong Li, Lizhu E, Jiayi Yang, Wenjing Liu, Mingliang Xu, Jianrong Ye

**Affiliations:** National Maize Improvement Center, Center for Crop Functional Genomics and Molecular Breeding, Department of Plant Genetics and Breeding, China Agricultural University, Beijing 100193, China

**Keywords:** dirigent protein ZmDRR206, maize kernel development, cell wall biogenesis, defense response, storage nutrient accumulation

## Abstract

Dirigent proteins (DIRs) contribute to plant fitness by dynamically reorganizing the cell wall and/or by generating defense compounds during plant growth, development, and interactions with environmental stresses. ZmDRR206 is a maize DIR, it plays a role in maintaining cell wall integrity during seedling growth and defense response in maize, but its role in regulating maize kernel development is unclear. Association analysis of candidate genes indicated that the natural variations of *ZmDRR206* were significantly associated with maize hundred-kernel weight (HKW). *ZmDRR206* plays a dominant role in storage nutrient accumulation in endosperm during maize kernel development, *ZmDRR206* overexpression resulted in small and shrunken maize kernel with significantly reduced starch content and significantly decreased HKW. Cytological characterization of the developing maize kernels revealed that *ZmDRR206* overexpression induced dysfunctional basal endosperm transfer layer (BETL) cells, which were shorter with less wall ingrowth, and defense response was constitutively activated in developing maize kernel at 15 and 18 DAP by *ZmDRR206* overexpression. The BETL-development-related genes and auxin signal-related genes were down-regulated, while cell wall biogenesis-related genes were up-regulated in developing BETL of the *ZmDRR206*-overexpressing kernel. Moreover, the developing *ZmDRR206*-overexpressing kernel had significantly reduced contents of the cell wall components such as cellulose and acid soluble lignin. These results suggest that ZmDRR206 may play a regulatory role in coordinating cell development, storage nutrient metabolism, and stress responses during maize kernel development through its role in cell wall biogenesis and defense response, and provides new insights into understanding the mechanisms of kernel development in maize.

## 1. Introduction

Maize (*Zea mays*) has a persistent endosperm that occupies the largest portion of the mature kernel by accumulating large amounts of nutrient reserves. Maize endosperm is a key determinant of kernel weight and grain yield. Maize kernel weight depends on the effective transport of carbohydrates from the maternal tissues to the filial tissues [1]. The transportation of nutrients from the maternal tissues to the central endosperm is dependent on several types of cells: the cells in the pedicel (PED) and the placento-chalazal region (PC), which are two key maternal tissues that control the supply of nutrients to filial tissues, as assimilates are unloaded from maternal vascular terminals in the PED, then pass the PC and the basal endosperm transfer layer (BETL) into the central endosperm through symplast and/or apoplast [2,3,4,5]. Seeds absorb maternal nutrients, including photo-assimilated sucrose, via transfer cells (TCs) at the maternal/filial interface. A typical maize endosperm at 6 days after pollination (DAP) contains BETL, which is a single layer of visibly elongated TCs with cell wall (CW) ingrowths, a embryo-surrounding region (ESR), starchy endosperm (SE), and an aleurone layer (AL) [4,6]. The BETL and ESR provide pathways for nutrient transport to the endosperm and embryo, respectively. The basal intermediate zone (BIZ) and the conducting zone (CZ) connect BETL with SE; SE and its central section, CSE, comprise large cells, are tissues functioning as a reservoir for starch and protein. As the outermost monolayer of cells, the AL stores vitamins and minerals, and provides signals to SE cells [4,7,8]. BETL cells function as the nutrient transporter, antimicrobic barrier, and signal mediator between filial and maternal tissues. There are three elemental factors induce the differentiation of BETL cells: sugar supply of maternal tissues, sugar demand of filial tissues, and requirement for defense against pathogens. The nutrient transport capacity of TCs depends on the unique CW ingrowths that create a high rate of transport flux. As a highly specialized form of AL, maize BETL cells play a major role in transferring nutrient solutes from the PC to the developing endosperm by facilitating nutrient transport through rib-like CW ingrowths to allow high levels of sugar transport from the maternal tissue, facilitate the development of endosperm cells, and provide the sugar needed for the higher sink strength of the embryo [4,9].

Maize kernel weight is remarkably influenced by the unloading of nutrients from the maternal tissue and their transportation from the BETL and BIZ to the upper part of the developing endosperm. The BETL cells produce extensive secondary wall ingrowths that increase the surface area of the plasma membrane (PM) and CW, to facilitate rapid nutrient uptake and transportation [10]. The reported BETL-specific genes can be classified into several groups, including genes encoding signal receptors/transducers, sugar converters/transporters, and lipid transporters [11]. The MYB-related protein, ZmMRP1, which is a master transcriptional regulator involved in BETL differentiation by regulating many well-known BETL-specific genes, including *ZmTCRR1* (*Transfer Cell Response Regulator1*) and *ZmTCRR2* that encode type-A response regulator class proteins and may involve in intercellular signal transduction [12,13]; *MEG1* [14], and *BETL1, BETL2, BETL9, and BETL10* [15,16]. *MEG1* encodes a small cysteine-rich secreted peptide (CRP) that acts as a BETL-specific key epigenetic factor and contributes to maternal nutrient translocation and partitioning in growing kernels [17,18]. BETLs are required for correct BETL development and nutrient allocation in developing kernels, they belong to the CRP family proteins that are secreted into surrounding pedicel tissue during plant pathogen responses [19]. ZmMRP1 and its target genes are essential for BETL function maintenance, and initiate the development of CW ingrowths in BETL cells at 6 to 9 DAP. BETL1 assists cell wall component deposition in the CW ingrowths and facilitate TC differentiation [20]. *BETL2* encodes a defensin-like protein to protect endosperm against pathogens [21]. BETL cells elongate and enormously increase their CW ingrowths at 12–15 DAP. The CW ingrowth development of BETL stops at 18–21 DAP, with a maximum density of CW ingrowths at 22–25 DAP, accompanying the declined expression levels of *BETL1* and *2* and *ZmTCRR1* and *2* [22].

Several factors have been reported to play important roles in regulating kernel development by influencing nutrient uptake and allocation in the maternal/filial interface region. A maize CW-bound invertase, Mn1 (Miniature seed1), is located in the PED and the BETL. Mn1 cleaves sucrose into hexoses to generate a physiological gradient of photosynthate, then the hexoses pass the PC through symplast and apoplast, and pass the BETL to the endosperm through apoplast [23]. Loss function of Mn1 results in smaller kernel size [1], while *Mn1* overexpression markedly increases kernel size and weight [24]. *ZmSWEET4c* encodes a sugar transporter that transports hexoses produced by Mn1 to the inner SE cells for starch synthesis or for the embryo development [1,25]. Mn1 and ZmSWEET4c are two kernel filling-related factors, their normal functions are required for the development of fully differentiated BETL with CW ingrowths [1,26]. ZmMRP1 is regulated by both the coordinated expression of *Mn1* and *ZmSWEET4c* and the hexose and auxin levels in BETL [1,6]. Sugars have dual functions as photosynthates (a nutrient) and regulatory signal molecules affecting the differentiation and development of endosperm cells. In maize, sugars also regulate cell fate in BETL via cross-talk with auxin, the altered sugar and hormone levels could affect BETL cell fate, resulting in reduced CW ingrowths in BETL of maize mutant *mn1-1* [27]. Auxin is important for the development of CW ingrowths and low auxin signaling could suppress the development of CW ingrowths in BETL cells [28]. *Defective18 (DE18)/ZmYUC1* encodes a rate-limiting enzyme for auxin synthesis. The mutation of *DE18* greatly decreases free IAA content in endosperm, impairs the development of CW ingrowths in BETL cells, and the expression of *DE18* is greatly affected by sucrose levels [29]. CW ingrowth formation enlarges the surface area of the PM and therefore requires abundant lipid precursors, including phosphatidic acid, triglyceride, and choline [30]. ZmCTLP1 is a choline transporter-like protein 1 that facilitates choline uptake, its mutation results in reduced choline content, aberrant development of CW ingrowths, and down-regulated expression of *Mn1* and *ZmSWEET4c*, thus leads to smaller kernels with a lower starch content [31]. The formation of CW ingrowth requires cellulose synthesis and deposition. ZmCESA5 encodes a cellulose synthase, the nonfunctional ZmCESA5 decreases nutrient uptake from maternal tissue, creating a degraded endosperm in the kernel, while *ZmCESA5* overexpression increases kernel weight [32].

Lignin is primarily associated with cell wall rigidity and strength, lignification is reported to associate with several plant traits, such as the cell wall structural integrity, secondary growth of vascular tissues, the strength of the stem and root, fungus resistance [33,34,35]. Maize BETL cells are critical for kernel development because of their extensive CW ingrowths that project several micrometers into the cytosol and maintain a consistent ultrastructure throughout kernel development [36]. The role of lignin incorporation in walls and ingrowths of BETL cells may be to stabilize such structure. BETL cells become lignified from early (6 DAP) to mid-development stages when undergoing active growth and the formation of CW ingrowths. It is reasonable to suggest that the inward growth of CW regions may require reinforcement by lignin, as this projection process must overcome the outward pressure of the living cell cytoplasm. Lignin may stabilize the ingrowth structure during its formation [36,37]. The lignification of starchy cell coincides with its growth from early to late developmental stages, and lignification does not inhibit growth of these cells. During the process of CW growth, lignin may possibly counterbalance various wall-loosening processes (mediated by growth-promoting agents), thus allowing the CW to expand without losing rigidity. The starchy cells accumulate large amounts of starch and protein starting at 12 to 14 DAP and continuing until physiological maturity, and they would undergo programed cell death during endosperm development [38,39]. Therefore, it is important for the starchy cell walls to be strong and flexible enough to endure such challenging changes, and lignin is such an important constituent to provide such abilities [35].

Maize Disease Resistance Response206 (ZmDRR206) plays a role in cell wall integrity (CWI) maintenance through its role in coordinate regulation of biosynthesis of cell wall components, *ZmDRR206* overexpression significantly increased disease resistance against *Fusarium graminearum* infection and drought tolerance in maize seedlings, while simultaneously compromised kernel development and seedling growth in maize [40]. Here, we showed that natural variations of *ZmDRR206* were significantly associated with hundred-kernel weight (HKW) by association analysis of candidate genes, and *ZmDRR206* overexpression in maize significantly decreased kernel size and HKW. *ZmDRR206* overexpression induced dysfunctional basal endosperm transfer layer (BETL), and down-regulated multiple BETL-development related, auxin biosynthesis and auxin signal-related genes in developing kernel; moreover, the developing *ZmDRR206*-overexpressing kernel had significantly reduced contents of cellulose and acid soluble lignin, compared to that of the WT kernel.

## 2. Results

### 2.1. Natural Variations of ZmDRR206 Are Significantly Associated with Maize Hundred-Kernel Weight

In order to determine the association of the natural SNPs of *ZmDRR206* with hundred-kernel weight (HKW), 513 maize inbred lines of tropical, subtropical, and temperate origins representing the global diversity of maize germplasm were used to identify SNPs [41]. Association analysis of candidate genes identified a total of 139 SNPs within a ±20 kb region of the *ZmDRR206* gene (that is the genomic region between the 20 kb up-stream of the start codon and the 20 kb down-stream of the stop codon of the target gene *ZmDRR206*), 80 of these SNPs were within the ±2 kb region of the *ZmDRR206* gene (Appendix A). The 102 SNPs were obtained from the genomic region within a ±5 kb region of the *ZmDRR206* gene by pairwise linkage disequilibrium calculation with R package IntAssoPlot, among them, 24 SNPs were identified within the coding region of the *ZmDRR206* gene, and one SNP within the coding region (chr2.S_243395225, v4) was identified to be significantly associated with variations in HKW (Figure 1). Therefore, we established that there was a natural SNP of *ZmDRR206* that was significantly associated with maize HKW.

### 2.2. ZmDRR206 Overexpression Affects Storage Products Accumulation in Maize Kernel

To investigate the role of *ZmDRR206* in maize kernel development, six independent *ZmDRR206*-overexpressing transgenic maize events were obtained from the c+enter for crop functional genomics and molecular breeding of China Agricultural University, and their T_4_ homozygous progenies were named as *DRR-OE*. *ZmDRR206*-overexpressing significantly decreased maize HKW and hundred-endosperm weight, while had no effect on maize hundred-embryo weight and kernel germination [40]. Unlike the fully filled, mature yellow kernel of the wild type (WT), *DRR-OE* kernels were opaque and small with pale color and shriveled appearance, especially at the bottom of the kernel. Such appearance was more severe on the ear of heterozygous *ZmDRR206* overexpressors, as the kernels had severely shriveled top, while their neighboring WT kernels grew better and bigger than other WT kernels that had no adjoining *DRR-OE* kernel (Figure 2A,B), indicating the WT kernels had stronger sink-strength than their neighboring *DRR-OE* kernels grew on the same ear. Cross and longitudinal dissection of the mature air-dried kernels showed that the *DRR-OE* endosperm was smaller with less vitreous endosperm and the *DRR-OE* kernel was easily broken, moreover, the lower part of the *DRR-OE* kernel was severely impaired with large cavities at the bottom and around the embryo, while the embryo showed no difference in appearance, compared to that of the WT kernels (Figure 2C)*,* indicating that the *DRR-OE* endosperm was more severely affected than the embryo by *ZmDRR206* overexpression. The aberrant endosperm with substantially reduced starch content is usually associated with reduced number of starch grain (SG) or smaller SG in diameter. Consistently, we found the starch content of the *DRR-OE* kernel was significantly decreased and the contents of total protein and lipid were significantly increased when these storage nutrients were measured according to equal biomass, compared to that in WT kernels (Figure 2D–F). The SG in the CSE of the mature dry maize kernels was checked under SEM, and the SG in the CSE of the *DRR-OE* kernel was significantly smaller in diameter, compared to that in WT kernel (Figure 2G,H). Thus, the *DRR-OE* kernel exhibits a reduced endosperm and unbalanced starch and protein accumulation. Furthermore, the *DRR-OE* kernels showed a higher content for most amino acids, except Cys and His, and the free proline content of *DRR-OE* was especially higher than that of the wild type (Figure 2I). Furthermore, we observed that *ZmDRR206* overexpression induced the mature transgenic Arabidopsis seeds to be slightly slender than that of the wild type (Appendix A) [40]. These data suggest that *ZmDRR206* plays an important role in storage nutrient metabolism during maize kernel development.

### 2.3. ZmDRR206 Acts as a Dominant Regulator for Storage Nutrient Accumulation in Maize Kernel

*ZmDRR206* overexpression resulted in remarkable changes in mature kernel appearance, this leads to the observation of the developing profile of the kernels in detail. The developing kernels of the homozygous *ZmDRR206* overexpressors were similar to WT at 15 DAP, they became distinguishable from the WT due to their light pigmentation beginning at 18 DAP (that is they appeared pale in color at 18 DAP), and they showed a dent or sunken appearance on the top of the kernel at 21 DAP. *ZmDRR206* had a dominant effect on maize HKW, as both the test-cross and back-cross between *ZmDRR206*-overexpresing plants and WT plants resulted in similar significant reduction in HKW and similar pale appearance in mature F1 kernel (Figure 3A–C), suggesting that *ZmDRR206* may be a dominant gene controlling endosperm traits by regulating storage nutrient accumulation in developing maize endosperm, similar to the usually observed effect of xenia on the endosperm traits of maize kernel controlled by the dominant genes contained in pollen. Moreover, the analysis with 200-kernel constant weight of the developing kernels revealed that the development profile of *DRR-OE* kernels differed significantly from WT kernels after 15 DAP, as their constant weight was always significantly smaller than that of WT kernels at every time point except 15 DAP (Figure 3D). These suggest that *ZmDRR206* plays a dominant role in storage nutrient accumulation in developing endosperm during maize kernel development.

### 2.4. ZmDRR206 Regulates the Development of the BETL and AL Cells in Maize Kernel

As the appearance of *DRR-OE* kernels differed from WT kernels at ~18 DAP, this prompted us to check cellular characterization of the developing kernels by paraffin sectioning. From the longitudinal section of the middle part of kernels, cytological difference could be observed: there were much less periodic acid-schiff (PAS)-stained granules accumulated in the CSE; the central part of ESR of *DRR-OE* kernel appeared empty with much less granules than that of the WT kernels, consistent with the greater sink strength of the embryo than that of the endosperm in developing maize kernel; and the embryo development was delayed in *DRR-OE* kernels, as the WT embryo had developed an obvious scutellum, coleoptile, two leaf primordia, and both shoot and root apical meristems, while the leaf primordia were delayed in *DRR-OE* kernels at 18 DAP (Figure 4A). These two kernels also displayed difference in phenotype of AL and BETL cells. The AL cell was a single, uniformly rectangular layer in the peripheral endosperm of the developing WT kernel; however, the AL cells of *DRR-OE* endosperm were thicker and variable in shape (short and round) and loosely arranged in *DRR-OE* kernels. Under TEM, the WT AL cell had many aleurone grains (AGs), while unexpectedly, the AL cells of *DRR-OE* kernel contained many starch grains (typical of SE cells), beside AGs. Additionally, the intercellular space between two AL cells was bigger and the cell wall was thicker in the *DRR-OE* kernel than that of the WT (Figure 4B,C). The TCs provide nutrient transport capacity through their unique CW ingrowths that increase the surface area of the plasma membrane, creating a high rate of transport flux. The shriveled endosperm is usually associated with aberrant BETL due to the possible impairment of critical transport functions associated with these cells during seed development. The BETL cells of WT kernel showed a characteristic slightly elongated shape with labyrinth-like CW ingrowths, while the BETL cells of *DRR-OE* were shorter in length with fewer thickenings or CW ingrowths. Moreover, the BIZ cells of WT kernel were extremely elongated with CW ingrowths, while the elongation of the BIZ cells of *DRR-OE* kernels was less pronounced, and these BIZ cells were short, enlarged or rounded similar to those of the starchy endosperm (Figure 4D). These results suggest that the *DRR-OE* endosperm develops dysfunctional nutrient TCs, especially the BETL cells with less wall ingrowth and short and round BIZ cells, and this may retard nutrient transport from the maternal tissue to endosperm. Consistently, the PAS-stained starch granules in CSE of the *DRR-OE* kernels appeared lighter in color, compared to that of WT kernels (Figure 4E), and the mature *DRR-OE* kernel was shriveled with significantly reduced starch content (Figure 2). These suggest that ZmDRR206 plays a role in regulation of the development of BETL and BIZ cells in maize kernel.

### 2.5. ZmDRR206 Alters the Expression of Defense-Related Genes in Developing Maize Kernel

To further investigate the molecular mechanism of *ZmDRR206* in regulation of maize kernel development, we obtained the transcriptome profiles of WT and *DRR-OE* developing kernel at 15 and 18 DAP by RNA sequencing. *ZmDRR206* overexpression induced 1252 differentially expressed genes (DEGs, 822 of them were up-regulated) in developing *DRR-OE* kernels at 15 DAP, while induced 1616 DEGs (791 of them were up-regulated) in 18 DAP *DRR-OE* kernels, using the thresholds of absolute fold change >2 and *p* < 0.05, compared to that in WT kernels (Appendix A). The Gene Ontology (GO) enrichment analysis (http://bioinfo.cau.edu.cn/agriGO/, accessed on 13 February 2020) of the DEGs from 15 DAP kernels of *DRR-OE* and WT showed that there were many DEGs encoding proteins with activity of oxidoreductase, monooxygenase, chitinase, and peroxidase; Moreover, these DEGs were enriched in biological process of defense-related functional categories, such as response to various stimulus (like hormone, biotic/abiotic stimulus), secondary metabolite biosynthesis and cell wall organization/biogenesis. Furthermore, large number of these DEGs were enriched in plant organ development-related functional categories, such as shoot, phyllome, tissue, xylem and meristem development, suggesting that *ZmDRR206* overexpression affected kernel development process at 15 DAP (Figure 5A). This is consistent with the observed developmental delay in embryo of *DRR-OE* kernels (Figure 4A). However, at 18 DAP, there are many DEGs encoding proteins with activity of oxidoreductase, chitinase, glucosidase, glucosyltransferase, UDP-glucosyltransferase, and beta-glucosidase. The latter four kinds of enzymes involve starch and sucrose metabolism and/or carbon metabolism, suggesting that these two metabolisms differed in the two kernels at 18 DAP. The GO enrichment analysis of the DEGs from 18 DAP kernels of *DRR-OE* and WT revealed that many DEGs were enriched in defense-related functional categories, such as response to various stimulus (like hormone, stress, biotic/abiotic stimulus), secondary metabolic process and defense response, while no organ development-related functional categories were enriched with DEGs from 18 DAP kernels of *DRR-OE* and WT (Appendix A; Figure 5B), indicating that there was no significant transcriptional difference in kernel development process between these two kernels at 18 DAP. Overall, there was no significant difference in nutrient reservoir activity between *DRR-OE* and WT kernels at 15 /18 DAP, as no enrichment of the DEGs from both the 15 and 18 DAP kernels was found to be enriched in this GO term. These results suggest that *ZmDRR206* overexpression significantly affected the expression of genes related to defense-related biological process in developing maize kernel at 15 /18 DAP, *ZmDRR206* may play a regulatory role in coordinating storage nutrient metabolism, cell development, and stress responses during maize kernel development.

### 2.6. The Kernel Development- or Starch Synthesis-Related Genes were Not Affected in Developing DRR-OE Kernels

However, *ZmDRR206* overexpression significantly decreased the starch content in mature maize kernel (Figure 2D) and delayed embryo development at 18 DAP (Figure 4A), we investigated the impact of *ZmDRR206* overexpression on the expression of the critical genes that play important role in kernel development and starch biosynthesis. Compared to the WT kernel, the expression level of *ZmDRR206* was significantly increased in the developing *DRR-OE* kernel at 15, 18 and 21 DAP, confirming the significant overexpression of *ZmDRR206* in the developing *DRR-OE* kernel. Furthermore, the auxin synthesis rate-limiting enzyme gene, *ZmYUC1*, was found to be significantly down-regulated in the developing *DRR-OE* kernel at 18 (0.362) and 21 DAP (0.407), compared to that in WT kernels. However, the expression of the kernel development regulator genes and the starch synthetic genes was not significantly different between these two kernels, including genes encoding bZIP17, O2, and chloroplast pyruvate phosphate dikinase1/2 (cyPPDK1/2), sucrose synthases (Sh1), AGPase subunit Sh2 and Brittle2 (Bt2), starch synthases (SSI, SSIIa, SSIII and GBSSI), confirmed by both the transcriptomic data and RT-qPCR analysis (Figure 5D; Appendix A). These indicate that the impact of *ZmDRR206* overexpression on kernel development and starch accumulation was not associated with the expression of the kernel-development-regulator and the starch synthetic genes. Contrasting to the significantly increased total protein content in *DRR-OE* relative to WT kernels (Figure 2F), the expression levels of almost all *zein* genes were down-regulated in developing *DRR-OE* relative to WT kernels, including the *Zein*s (15 kD, 19 kD) that confirmed by RT-qPCR (Appendix A). When the contents of zein protein and total protein were measured according to equal biomass, the percentage of zein protein to total protein was slightly decreased in *DRR-OE* relative to WT kernel, suggesting that the non-zein protein content increased. Accordingly, the expression of multiple non-zein storage protein genes was up-regulated in developing *DRR-OE* relative to WT kernel, including five *globulin* genes, two *germin-like protein* (*GLP*) genes and five *late-embryogenesis protein* (*LEA*) genes, while the *2S albumin* genes did not differ between the two kernels (Appendix A), suggesting that additional protein storage pathways were activated in the developing *DRR-OE* kernel, compared to that of WT. These results suggest that the dysfunctional endosperm development of *DRR-OE* kernel was not associated with the kernel-development-regulator or starch synthetic activity.

### 2.7. The BETL Development- and Auxin Signal-Related Genes were Down-Regulated in Developing BETL of DRR-OE Kernel

Maize kernel development is directly influenced by the unloading of nutrients from the maternal tissues and their passage through the transfer tissue of the BETL and the BIZ to the upper part of the endosperm. *ZmDRR206* affects storage nutrient accumulation in maize kernel, but has no significant effect on the expression of starch synthetic enzyme genes, while the BETL cells of *DRR-OE* kernel were shorter with less wall ingrowth, the BIZ cells were short and round. This prompts us to check transcriptomic profile of the BETL-tissue in developing maize kernels. *ZmDRR206* overexpression induced 979 DEGs in BETL tissue of *DRR-OE* kernels at 18 DAP, 619 of these DEGs were down-regulated, compared to that in WT kernel (Appendix A). GO analysis of these down-regulated DEGs revealed that many of them encoding proteins with molecular function of monooxygenase (22), carboxylic acid transmembrane transporter (12), chitinase (8), transmembrane transporter (52, including the transporters for sugar and ion), transporter activity (59), etc. That is, many BETL and its neighboring tissue (including PED and PC) located transporter genes, which encoding for transporters for ions, amino acids and sugars, were found to be significantly down-regulated in the BETL of *DRR-OE* relative to WT kernel. Additionally, these down-regulated DEGs were significantly enriched in chitin catabolic process and transmembrane transportation-related functional categories, including transportation of amino acid, metal ion, sugar and ion (Figure 6A,B). In fact, these included 33 ion transporters, 12 sugar transporters, 14 amino acid transporters, and two lipid transporters (Appendix A). The expression of genes encoding important regulators of BETL development, including *ZmMRP1*, *Mn1*, *ZmSWEET4c*, *ZmTCRR1*, *MEG1, BETL3, BETL9,* and *BETL10*, together with a few genes associated with BIZ cell development, were all down-regulated in BETL of *DRR-OE* kernel, most of them were significantly down-regulated, compared to that of WT (Figure 6C). Auxin plays a major role in the development of CW ingrowths of BETL cells by regulating cell expansion through activation of cell wall synthesis and modification-related genes. The expression levels of the auxin biosynthesis (*ZmYUC*s, including *ZmYUC1/de18*) and auxin signal-related genes, including genes encoding SMALL AUXIN UP RNA12 (SAUR12), IAA27, AUX/IAA, indole-3-acetic acid-amido synthetases (GH3s), indole-3-acetate beta-glucosyltransferases (IAGs), auxin-induced proteins (AIP) and auxin responsive protein (ARP), were all significantly down-regulated in BETL tissue of *DRR-OE* kernel, compared to that of WT (Figure 6D), indicating auxin synthesis and signal were down-regulated in BETL of *DRR-OE* kernel. Furthermore, some cellular transportation and cell wall modification-related genes, including genes encoding bidirectional sugar transporter sweet13/14 (BSTS), sucrose transporter (SUT), UDP-glycosyltransferase (UGT), beta-glucanase (BG), chitinase (CHN), were all down-regulated (Appendix A; Figure 6E). The down-regulated expression of some of these genes were further confirmed by RT-qPCR, including *ZmYUC1, ZmMRP1*, *ZmMn1, ZmTCRR1*, *ZmMEG1* and *ZmBETL9* (Figure 6F). These results suggest that *ZmDRR206* overexpression suppressed the development and functional activity of BETL cells in developing kernel, *ZmDRR206* plays an important role in BETL development.

### 2.8. The Cell Wall Organization/Biogenesis was Altered in the Developing BETL of DRR-OE Kernels

GO analysis revealed that many of the up-regulated DEGs (induced by *ZmDRR206* overexpression in BETL tissue of developing kernel) encoding proteins with activity of oxidoreductase, cellulose synthase, acetyltransferase, peroxidase, UDP-glycosyltransferase, UDP-glucosyltransferase, etc. Additionally, these up-regulated DEGs were significantly enriched in cell wall organization/biogenesis, secondary cell wall biogenesis, polysaccharide biosynthetic process, carbohydrate metabolic process, glucan biosynthetic process, xylan biosynthetic process, hemicellulose metabolic process, cell wall polysaccharide metabolic process, oxidation-reduction process, defense response to fungus, cell wall modification, suggesting enhanced metabolism in these processes in *DRR-OE* BETL cells (Figure 7A). The cell wall biogenesis-related genes, including *ZmCesA10, 11, 12*, *7*, *xyloglucan endotransglucosylase hydrolase* (*XET*) and five *expansin* (*EXP*) genes, were also up-regulated in developing BETL of *DRR-OE* kernel (Figure 7B), the up-regulated expression of *ZmCesA10* was also confirmed by RT-qPCR (Figure 6F). However, the contents of cellulose and acid soluble lignin (ASL, the main lignin in cell wall) were significantly decreased, while the content of acid insoluble lignin (AIL, the main lignin in middle lamella) was significantly increased and the content of semi-cellulose was slightly increased without significance, while the total lignin content was similar in the developing *DRR-OE* relative to the WT kernel at 18 DAP (Figure 7C). Moreover, ion channels constantly monitor the state of the cell wall and initiate adaptive changes in both cellular and cell wall metabolisms [42]. Contrasting to the down-regulated DEGs that encode ion transporter in developing BETL of *DRR-OE* kernel (Figure 6A,B), the contents of mineral elements measured according to equal biomass, especially the contents of potassium (K), phosphorus (P), iron (Fe) and zinc (Zn) were all significantly increased in the mature *DRR-OE* kernels, while the content of sodium (Na) was significantly decreased and the content of magnesium (Mg) was elevated without significance, compared to that in WT kernels (Figure 7D,E), indicating that cellular osmotic conditions (ion contents) are altered during *DRR-OE* kernel development. These results indicate that ZmDRR206 involves in regulation of cell wall organization/biogenesis in the BETL during maize kernel development.

## 3. Discussions

### 3.1. ZmDRR206 Plays a Role in Regulation of Endosperm Development in Maize Kernel

The developmental process of seed is sensitive to environmental abiotic and biotic stresses, as the developing young seed is very vulnerable. *Disease Resistance Response206* (*DRR206*) is identified to be an inducible pathogenesis-related gene, involved in lignin/lignan biosynthesis. Lignan has antifungal properties, while lignin is an important part of the structural protection as well as defense [43,44]. ZmDRR206 involves in maintenance of cell wall integrity through its role in coordinate regulation of biosynthesis of cell wall components during maize seedling growth and interaction with the environment [40]. Here, the significant effects of *ZmDRR206* on kernel development were first demonstrated by association analysis of candidate genes [41], the natural SNPs of *ZmDRR206* were significantly associated with maize HKW (Figure 1, Appendix A). ZmDRR206 plays an important role in maize endosperm development by regulating the development of TCs in BETL and BIZ, as *ZmDRR206* overexpression induced maize kernel to be small with defective endosperm and significantly reduced starch content (Figure 2), and the TCs in BETL were short with less CW ingrowth and the TCs in BIZ were short and round in *DRR-OE* maize kernel (Figure 4). These are consistent with the relatively higher expression level of *ZmDRR206* in BETL, ESR, AL, and CZ in developing maize kernel, based on analysis of the published RNA-seq datasets [45]. Both test-cross and back-cross between *ZmDRR206* overexpressor and WT induced significantly decreased HKW (Figure 3), further suggesting that *ZmDRR206* acted as a dominant gene controlling endosperm traits in maize, similar to the effect of xenia on the starch content of maize kernel as pollen usually contains dominant genes controlling endosperm traits [46].

*ZmDRR206* did not affect embryo development, *DRR-OE* kernel had similar germination rate as the WT, the sink strength of developing embryo was stronger relative to the developing endosperm, thus causing the almost empty ESR region in the developing kernel (Figure 4A) and big cavity around the embryo in the mature *DRR-OE* kernel (Figure 2C). The embryo is the major site for the storage of lipid and protein in maize kernel [47]. The significantly reduced endosperm and the unchanged embryo of mature *DRR-OE* kernel resulted in the significantly higher ratio of embryo: endosperm, this might contribute to the significantly increased contents of lipid and total protein in the mature *DRR-OE* relative to WT kernel. Moreover, the increased protein content might also associate with the up-regulated expression of non-zein protein genes such as globulin, GLP and LEA in the *DRR-OE* kernels, which were all primarily expressed in maize embryo. Minerals (including iron) are stored in the AL and embryo during maize kernel development. The AL also provides signals to pathogen defense at maturity [48]. The enlarged Al cells and the unchanged embryo of *DRR-OE* kernel might account for the increased contents of the minerals (K, P, Fe and Zn) in *DRR-OE* relative to WT kernel. These data suggest that *ZmDRR206* may play a regulatory role in coordinating cell development and storage nutrient metabolism during maize kernel development.

### 3.2. ZmDRR206 Regulates Maize Endosperm Development through Its Role in Cell Wall Biogenesis and Defense Response

Experimental results confirmed the presence of lignin in developing BETL and starchy cells throughout the cell growth period in maize kernel. BETL cells have extensive inward wall projections, while the starchy CWs are much thinner. The formation of the rib-like CW ingrowth in the TCs of cereal seeds requires synthesis and deposition of cell wall components, including cellulose and lignin [32,35,36]. Lignin may act as a ubiquitous structural molecule to stabilize the ingrowth structure during its formation by reinforcing the inward growth of CW regions, thus overcoming the outward pressure of the living cell cytoplasm. Although the starchy cells grew even more significantly than the BETL cells, these cells also contain lignin throughout development [36,37]. The lignification of maize endosperm CWs is important for maintaining the CWI and strength, as the starchy cells accumulate large amounts of starch and protein, while they have to withstand severe desiccation (most water would be lost in these cells at late development stages) and maintain their tightly arranged starch granules and protein bodies [38]. Maize mutants with shriveled endosperm and kernel abnormalities were usually associated with the dysfunction of the CW ingrowths (the impairment of critical transport functions of these TC cells) in the developing BETL [1,26,32,49]. CW ingrowths increase the PM surface area of BETL cells and allow high levels of sugar transport from the maternal plant to the kernel, facilitate the development of endosperm cells and provide the sugar needed to meet the higher sink strength of the embryo. The levels of monosaccharides are rate-limiting factors for CW polysaccharide synthesis and glycosylation reactions in the BETL, the *mn1* mutant with dysfunctional CW ingrowths is short of free apoplastic hexose in the BETL due to its disability to cleave incoming sucrose [27]. As the cell′s energy factories, mitochondrial activity is functionally related with BETL development: large amounts of mitochondria in the dense cytoplasm of BETL cells are typically close to the CW ingrowths [26]; maize Dek35 is a nucleus-encoded mitochondrial pentatricopeptide repeat (PPR) protein that is essential for the *cis*-splicing of mitochondrial *nad4* intron 1, the *dek35* mutants have poorly developed BETLs with weakly formed CW ingrowths and their kernels are severely defective [50]. Cellulose synthesis and deposition are important for the formation of the CW ingrowths of BETL cells. *ENB1/ZmCESA5* is responsible for the CW ingrowth formation in BETL cells of developing maize kernel, the kernel of maize *ZmCESA5* mutant was small and shrunken due to dysfunctional BETL [32]. Similarly, *ZmDRR206* overexpression also remarkably reduced CW ingrowth formation in BETL cells of developing kernel and ultimately caused small and shriveled kernel with significantly reduced starch content (Figure 2). Additionally, the significantly reduced contents of cellulose and ASL in *DRR-OE* kernels were consistent with the remarkably reduced CW ingrowth formation in BETL cells of developing *DRR-OE* kernel (Figure 4D and Figure 7C). These indicate that ZmDRR206 plays an important role in regulation of the BETL development in maize kernel.

Both sugar and auxin play important roles in formation of CW ingrowths in developing BETL cells. The accumulation of free auxin in the BETL cells required for kernel development is depending on *ZmYUC1* expression, which can be activated by the release of sucrose from the maternal tissue to the developing kernel, and causes the response of local auxin levels to sucrose changes in the endosperm [27]. Low sucrose level would suppress *ZmYUC1* expression and cause low auxin level in the endosperm, and low auxin signaling could further suppress the development of flange ingrowths in BETL cells [27,28]. Sucrose induces the expression of *ZmCESA5* and *SWEET4c*, promotes the synthesis of CW ingrowths, which transport more sucrose from the maternal plant to the kernel to induce their expression [1,32]. Such reinforced regulation between sucrose and CW ingrowth development ultimately contribute to grain filling by establishing highly developed CW ingrowths in BETL cells. Consistent with the observed reduced CW ingrowth formation in BETL cells and the short, round phenotype of BIZ cells of *DRR-OE* kernel, the expression of genes encoding the important regulators of BETL development, including *ZmMRP1*, *Mn1*, *SWEET4c*, *ZmTCRR1*, *MEG1* and *BETL3, BETL9, BETL10*, and BIZ cell development-related genes were down-regulated in BETL of *DRR-OE* kernel, compared to that of WT (Figure 6C,F). Auxin plays a major role in the development of CW ingrowths by regulating cell expansion through activation of cell wall synthesis and modification-related genes. The expression levels of auxin biosynthesis and auxin signal-related genes, including *ZmYUC*1, *SAUR*, *GH3*s, *IAA27*, *AUX/IAA*, *IAG*s, and a few *EXP*s, were down-regulated in BETL tissue of *DRR-OE* kernel, compared to that of WT (Figure 6D,F). Similarly, the important regulators of BETL development and auxin signal genes were also down-regulated in the developing kernel of nonfunctional *ZmCESA5* mutant [32]. Additionally, the *DRR-OE* kernels showed more severely shrunken appearance on the heterozygous ear of the *ZmDRR206* overexpressors, and their neighboring WT kernels were rounder and larger than other WT kernels without adjoining *DRR-OE* kernel (Figure 1A), indicating the sink-strength of *DRR-OE* kernels were weaker than their neighboring WT kernels grew on the same ear, and less sucrose from the maternal tissue was transported to the developing *DRR-OE* kernel than that of WT kernel, thus resulted in the lower expression level of *ZmYUC1* in developing *DRR-OE* kernel than that of WT kernel, *ZmYUC1* level in developing *DRR-OR* kernel was significantly decreased, which was 0.362 and 0.406 of that in WT kernel at 18 and 21 DAP, respectively (Figure 5D). These results suggest that *ZmDRR206* overexpression might reduce sucrose transportation to developing kernel and thus suppress the development and functional activity of BETL cells in developing kernel, *ZmDRR206* has a profound role in maize endosperm development.

The cereal endosperm specific basic helixloop–helix (bHLH) transcriptional factor (TF), O11, functions as a central regulator in the gene networks governing maize endosperm development, nutrient metabolism, and stress response. O11 directly regulates many stress response genes [51]. *ZmDRR206* was found to be increased to a fold-change of 10.83 (*p*-value = 4.80 × 10^−3^) in 15 DAP endosperm of *o11* to that of WT. Accordingly, large amount of DEGs were significantly enriched in defense response- and biosynthesis of secondary metabolite-related functional categories in the transcriptomic data in *DRR-OE* relative to WT kernel at 15 and 18 DAP (Figure 5A,B), indicating defense response-related biological processes were constitutively activated in developing *DRR-OE* kernel. Moreover, many up-regulated DEGs induced by *ZmDRR206* overexpression in BETL tissue of developing kernel were significantly enriched in cell wall organization/biogenesis-related functional categories, indicating that these biological processes were altered in developing *DRR-OE* maize kernel. The secondary cell wall biogenesis-related genes, including *ZmCesA10,* were up-regulated in developing BETL tissue of *DRR-OE* kernel (Figure 6 and Figure 7). *ZmDRR206* overexpression altered synthesis of cell wall components, the contents of cellulose and ASL were significantly reduced, while the content of AIL was increased in developing *DRR-OE* relative to WT kernel at 18 DAP. The cell wall lignin was mainly composed of ASL, while the lignin in intercellular layer (middle lamella) was mainly AIL (Klason lignin). The reduced cellulose and ASL might associate with the reduced formation of CW ingrowths in BETL cells of developing *DRR-OE* kernel, and ultimately caused defective endosperm in *DRR-OE* kernel (Figure 2, Figure 3 and Figure 4); the increased AIL might associate with enhanced or constitutively activated defense response in developing *DRR-OE* kernel. The constitutively activated defense response-related biological processes thus enabled more energy supply to defense and less to growth, and disabled the energy-consuming process of endosperm development. However, the expression of most cellulose synthase encoding genes was not different in developing kernels/BETLs between *DRR-OE* and WT, except the *CesA7*, *10*, *11*, *12* with low expression levels in BETLs (the expression of *ZmCESA5* was 1.139 in *DRR-OE* relative to WT kernel at 18 DAP). The reduced cellulose content in 18 DAP *DRR-OE* kernel might associate with the constitutively activated defense response and the insufficient energy supply (the reduced levels of monosaccharides). The measurement of sugar and auxin levels in developing BETLs and endosperms of *DRR-OE* and WT kernels would provide more evidence for the role of *ZmDRR206* in maize kernel development. All these data suggest that *ZmDRR206* may play a regulatory role in coordinating cell development, storage nutrient metabolism, and stress responses during maize kernel development.

## 4. Materials and Methods

### 4.1. Association Analysis

For association analysis of *Zea mays* candidate genes to identify *ZmDRR206* variants related to HKW, we downloaded phenotypic data for 100-kernel weight of 513 diverse maize inbred lines [41] and the original genotypic data of 540 diverse maize inbred lines including approximately 2.65 million SNPs integrated from RNA-seq, 50 K array, 600 K and GBS technology [52] from MaizeGo (http://www.maizego.org/, accessed on 8 May 2022). These lines have tropical, subtropical, and temperate origins and hence represent the global diversity of maize germplasm. Genotyping data with minor allele frequency (MAF) below 0.05 were filtered then to implement genome-wide association analysis with a linear mixed model where the first three principal components and kinship matrix were used to control population structure and polygenic effect, respectively, in R package GAPIT (v3) [53]. 102 SNPs within the 5 kb upper distance from the start codon and the 5 kb down distance from the stop codon of the target gene *ZmDRR206* were used to calculate the pairwise linkage disequilibrium in R package IntAssoPlot [54]. A Bonferroni-corrected significance threshold (*p* ≤ 0.01/102 = 9.803922 × 10^−5^, −log_10_ (*p*) = 4.00) was used to identify the significant casual SNPs within the gene *ZmDRR206*.

### 4.2. Plant Materials

The transgenic maize plants that contained the construct *pUbiquitin*: *ZmDRR206*, in which *ZmDRR206* was driven by the maize *Ubiquitin* promoter, were obtained by *Agrobacterium* mediated genetic transformation with the immature embryos of the *Zea mays L*. LH244 inbred line (WT, used as the control in the afterward experiments), done by Center for Crop Functional Genomics and Molecular Breeding, China Agricultural University. Six independent *ZmDRR206* overexpression maize transgenic events and their T_4_ homozygous progenies were developed by continuous selfing, genotyping and named as *DRR-OE* for the following experiments. In summer, the maize plants were cultured in Shangzhuang experimental station, Beijing of China (39°54′ N, 116°25′ E); and in winter, they were grown in Sanya, Hainan of China (18°30′ N, 108°81′ E) every year.

### 4.3. Cytological Observations

Kernels at 18 DAP were freshly collected and cut longitudinally into three equal parts. The central part containing the intact embryo was selected to fix with formalin–acetic acid–alcohol (FAA) solution (50% ethanol, 10% glacial acetic acid, 35% ddH_2_O, and 5 mL 37% formaldehyde of 100 mL FAA solution) for 12–14 h at 4 °C. The fixed material was dehydrated in a gradient ethanol (50%,70%, 85%, 95%, and 100% ethanol in H_2_O [*v*/*v*]), which was subsequently replaced with xylene, infiltrated and embedded in Paraplast Plus (Sigma-Aldrich, St. Louis, MO, USA; cat. no. P3683). The samples were cut into 10 μm sections. The paraffin sections were stained with periodic acid-schiff stain (PAS) and finally imaged with the digital microscope (Olympus BX51). For TEM analysis, the developing kernels were freshly collected at 18 DAP and small pieces of the AL adjacent tissue, containing pericarp, AL and ~4 layers of cell adjacent to AL, were fixed with 2.5% glutaraldehyde and 1% osmium tetroxide at 4 °C for 12 h. The samples were subsequently washed several times with phosphate-buffered saline (PBS; pH 7.2). After dehydration, they were transferred to propylene epoxide and gradually infiltrated with acrylicresin. Ultrathin sections of the samples were cut with a diamond knife and photomicrographs, then observed using the transmission electron microscope H7600 (Hitachi, Tokyo, Japan). For scanning electron microscopy (SEM) analysis of starch grains in CSE cells, the air-dried mature kernels were collected and cut longitudinally at the middle. The samples were vacuum dried, spray coated with gold, and observed using a scanning electron microscope TM4000 Plus (Hitachi, Tokyo, Japan).

### 4.4. Phenotypic Analysis of Maize

For the phenotypic analysis of *ZmDRR206* overexpressors, fully filled mature, air-dried kernels from the middle part of the well-pollinated ear were used for measuring HKW. The time-course measurement of constant weight of developing kernels of wild-type and *ZmDRR206*-overexpressing lines were collected at 15, 18, 21, 24, 27, 30 DAP and mature kernels, each replicate was consisted of 40–50 kernels from the middle part of the ear and three to five ears were used per genotype. Each time point had three replicates per genotype.

### 4.5. Measurement of Starch, Protein and Lipids

The mature WT or *DRR-OE* kernels were collected from different well-filled ears, kernels from the middle part of the same ear were pooled as a single replicate, and mature kernels for each sample were ground into a fine powder in liquid nitrogen and flour was weighed for analysis. Three biological replicates were used for the subsequent analysis. The measurement of total protein and zein protein, lipid, and amino acid contents were measured according to a previously described protocol [55]. A 50 mg flour sample was incubated overnight in 1 mL lysis buffer (12.5 mM sodium borate, 1% SDS, 2% β-mercaptoethanol, 1% cocktail (Merck, Germany), and 1% phenylmethylsulfonyl fluoride) in a 37 °C shaker. The mixture was centrifuged for 10 min at 12,000 rpm, the supernatant was carefully transferred into a new 1.5-mL centrifuge tube. The total protein was measured according to a BCA standard kit (Thermo Fisher Scientific, MA, USA). For total starch measurements, the obtained powders were dried to a constant weight and the starch quantification was performed followed the method described by instructions of amyloglucosidase/α-amylase starch assay kit (Megazyme), as described by Feng et al. [51]. For the lipid measurements, 100 mg of dried flour was used for lipid extraction and then measured according to a previously described protocol [55]. Each assay above was independently performed three times on the kernels collected from three different ears.

### 4.6. RNA-seq and qRT-PCR Assays

The developing maize kernels were collected at 15, 18, 21 DAP from the middle part of three to five well-grown ears and were cut into the mixing pool at each time point, and then frozen in liquid nitrogen, followed by RNA extraction. The BETL-tissue used for RNA-sequence was selected from the developing WT/*DRR-OE* maize kernels by cutting off the bottom tissue of the young kernel from multiple well-grown ears, that is, the tissues might contain the BETL and its adjacent cells, including the PED, PC and BIZ cells. Total RNA was isolated using TRIzol reagent (Biotopped) and RNA integrity was evaluated using a Bioanalyzer 2100 (Agilent). The libraries were sequenced on an illunima nova 6000 at Berry Genomics (Beijing) and 150 bp paired-end reads were generated. Two independent replicates were performed. Raw data were firstly subjected to quality control using FastQC (v.0.11.9). Then, reads were mapped to the Maize genome (B73 RefGen_v4, AGPv4) using HISAT2 (v.2.2.0) with default parameters. FPKM of each gene was calculated using Cufflinks (v.2.2.1). The differential expression analysis was performed and significant DEGs were identified as those with a *p* value (one-way ANOVA test) of differential expression above the threshold (|log2| > 1, *p* < 0.05). GO enrichment was performed with the accession numbers of significant DEGs via agriGO v2.0.

Real-time RT-qPCR was used to check the relative expression level of the kernel development-related and other genes that showed significantly different fold changes in expression level between the *DRR-OE* and WT. For RT-qPCR experiments, the developing kernels and the BETL tissues sampled as described above were used for total RNA extraction and used for first-strand cDNA synthesis, using approximately 1 µg of total RNA and RT Master Mix with gDNA Remover (Takara Bio), which contains reverse transcriptase, oligo (dT) primer and random hexamer primer. RT-qPCR analyses were conducted using PowerUp^TM^ SYBR Green Master Mix (Applied Biosystems, Carlsbad, CA, USA) on an ABI 7500 thermocycler (Applied Biosystems). The RT–qPCR amplification reactions were performed in a total volume of 20 µL for each reaction, which contained 10 µL SYBR Green Realtime PCR Master Mix, 1 µL forward and reverse primers (10 µM), 1 µL cDNA dilution products and 8 µL H_2_O. The maize *GAPDH* (accession no. X07156) and *Ubiqutin* genes were employed as the endogenous control, the sequence of all the use primers was listed in Appendix A. The PCR was performed with the following conditions: 94 °C for 2 min; 40 cycles of 94 °C for 30 s, 60 °C for 30 s, and 72 °C for 30 s.

### 4.7. Analysis the Contents of Cell Wall Components in Developing Maize Kernels

The developing maize kernels of *DRR-OE* and WT at 18 DAP were sampled, dried and homogenized to a fine powder using a mixer mill at 25 Hz for five minutes. The powdered kernels were prepared to exclude protein and other UV-absorbing materials done according to the description by Moreira-Vilar et al. [56] and sequentially used for measuring the contents of the main cell wall components according to the description by Zhong et al. [40].

### 4.8. Measurement of the Mineral Element Content

To measure the contents of the mineral elements in maize mature kernels, including Na, K, P, Mg, Zn and Fe, a microwave digestion method was used. The WT and *DRR-OE* kernels were dried at 60 °C to constant weight for 24 h. The kernels were homogenized to a fine powder using a mixer mill at 25 Hz for five minutes and the dry weights of the powder were measured. Approximately 0.5 g kernel powders were digested with HNO_3_ at room temperature overnight. The digested solution was then diluted with deionized water and analyzed for the concentrations of the mineral elements, using atomic absorption spectrophotometry (Thermo Elemental *ICAP6300*). The contents were then calculated with the concentrations of the mineral elements in the diluted solution and tissue dry weights.

### 4.9. Statistical Analysis

The statistical analysis was conducted using Student’s *t*-test between the WT and *DRR-OE* kernels to determine statistical significance in independent experiments. The statistical significance was accepted at *p* < 0.05 (*), *p* < 0.01 (**) and *p* < 0.001 (***) for various kernel or other parameters. All data with multiple levels were tested by multiple comparisons using the least significance difference (LSD) test method at the significance level of 0.05.

## 5. Conclusions

In this paper, we investigated the biological role of a maize DIR, ZmDRR206, in regulation of kernel development. Gene association analysis of diverse maize germplasm revealed that significant SNPs associated with HKW were detected in the ZmDRR206 gene region. *ZmDRR206* plays a dominant role in storage nutrient accumulation in developing endosperm due to its role in cell wall biogenesis and defense response. Its overexpression resulted in dysfunctional BETL with less CW ingrowth, down-regulated BETL-development- and auxin signal-related genes, also significantly reduced contents of cellulose and acid soluble lignin (the main lignin in cell wall) and constitutively up-regulated defense response in developing maize kernels. These transcriptional and physiological changes ultimately led the mature *DRR-OE* maize kernel to be small, opaque and shrunken with significantly reduced starch content and significantly decreased kernel size and HKW. Although* ZmDRR206* overexpression increased disease resistance, and enhanced drought tolerance in maize, it also resulted in a small kernel and diminished seedling growth, the same negative effects as other disease resistance genes. To alleviate or eliminate the growth suppression side-effect of *ZmDRR206* overexpression on maize seedling growth and kernel development, it would be beneficial to put *ZmDRR206* under the control of an inducible promoter to make possible the expression of *ZmDRR206* induced to increase rapidly upon pathogen infection or drought/heat stress, while keeping basal expression under normal growth conditions for adaptation to the changing climate.

## Figures and Tables

**Figure 1 ijms-24-08735-f001:**
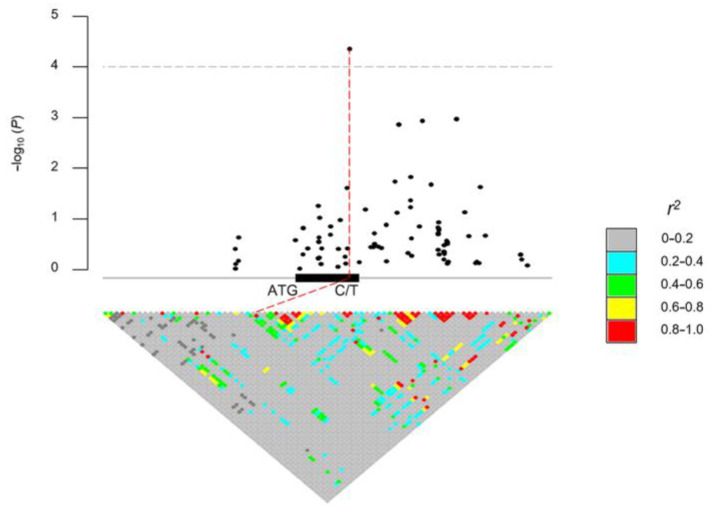
Association of SNPs within a ±5 kb region of the *ZmDRR206* gene with hundred-kernel weight among 513 maize inbred lines. The pairwise linkage disequilibrium was calculated in R package IntAssoPlot with the 102 SNPs obtained from the genomic region between the 5 kb up-stream of the start codon and the 5 kb down-stream of the stop codon of the candidate gene *ZmDRR206*. The significant casual SNP (C/T), chr2.S_243395225 (v4), within *ZmDRR206* gene is shown in red line, together with its chromosome loci and significance (4.36). *r^2^
*is linkage disequilibrium.

**Figure 2 ijms-24-08735-f002:**
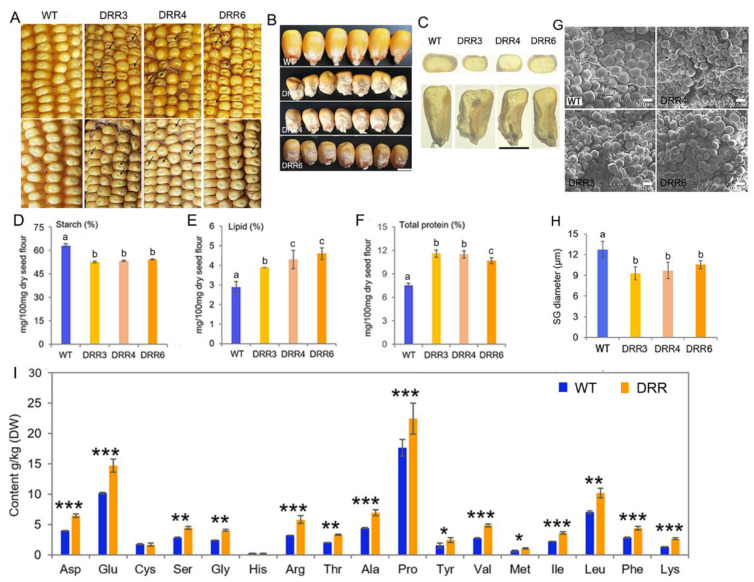
The kernel characterization of the transgenic *ZmDRR206* overexpressors. (**A**): Comparison of kernels on the mature ears of WT and *ZmDRR206* overexpressor (*DRR*-*OE*). The severely shrunken kernels in pale color (the dark arrows indicate) on the ears of the heterozygous (upper, the segregating ear) and the top-indented kernels in pale color (the dark arrows indicate) on homozygous ear (lower) of *ZmDRR206* overexpressors were presented, respectively. (**B**): The phenotype comparison of the mature WT and *DRR*-*OE* kernels. (**C**): Comparison of the cross and longitudinal sections of mature WT and *DRR*-*OE* kernels. The *DRR*-*OE* kernels had less vitreous endosperm and large cavity around the embryo and at the bottom of the kernels. Bar = 5 mm in (**B**,**C**). (**D**): The starch content of *DRR*-*OE* kernel was significantly less than that of the WT kernel. (**E**,**F**): The contents of lipid (**E**) and the total protein (**F**) was significantly increased in mature *DRR-OE* kernels, compared to that in WT kernels. Different lowercase letters (a, b, c) above the bars indicate significant differences based on a one-way analysis of variance (ANOVA) analysis followed by multiple comparisons using the least significance difference (LSD) test (*p* = 0.05). Error bars indicate the mean values ± SD of three replicates per genotype. (**G**): The phenotype of the starch granule in the CSE of the mature kernel by scanning electron microscope. Bar = 10 um. (**H**): The diameter of the starch granule was significantly smaller in *ZmDRR206*-overexpresing kernels than that of the WT kernel. (**I**): The amino acid content analysis in the mature kernel of WT and *DRR-OE*. DRR were the average of the three transgenic everts *DRR-OE3*, *DRR-OE4* and *DRR-OE6*. WT is the wild-type inbred line LH244; DRR3, DRR4 and DRR6 are the *ZmDRR206* overexpressing transgenic everts *DRR-OE3*, *DRR-OE4* and *DRR-OE6*. Data are presented as the mean values ± SD of three replicates per genotype. The asterisks represent a significant difference at * *p* < 0.05, ** *p* < 0.01, and *** *p* <0.001, according to a paired Student’s *t*-test.

**Figure 3 ijms-24-08735-f003:**
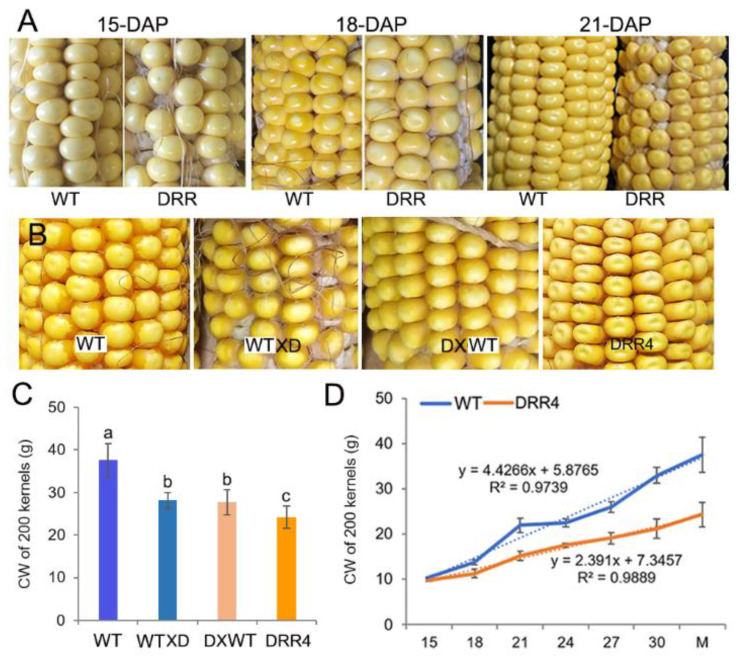
The dominant effect of *ZmDRR206* on kernel development. (**A**): The phenotype of the developing maize kernels. The developing kernels of *ZmDRR206* overexpressors were similar to WT at 15 days after pollination (DAP), while they appeared pale in color at 18 DAP and showed a dent or sunken appearance on the top of the kernel at 21 DAP. (**B**): The kernel phenotype of the test-cross and back-cross between *ZmDRR206* overexpressor (*DRR4*) and WT. (**C**): The constant weight (CW) of 200 mature maize kernels. WT X D is to pollinate pollens from *DRR4* plant on WT ear; D X WT is to pollinate pollens from WT plant on *DRR4* ear. Different lowercase letters (a, b, c) above the bars indicate significant differences based on ANOVA analysis followed by multiple comparisons using the LSD test (*p* = 0.05). Error bars indicate the mean values ± SD of three replicates per genotype. (**D**): The development profile of the maize kernel by constant weight analysis of 200 kernels. WT is the wild-type inbred line LH244, *DRR4* is *ZmDRR206* overexpressing transgenic evert *DRR-OE4*. The numbers in *x*-axis are the days after pollination, M is the harvested mature kernel.

**Figure 4 ijms-24-08735-f004:**
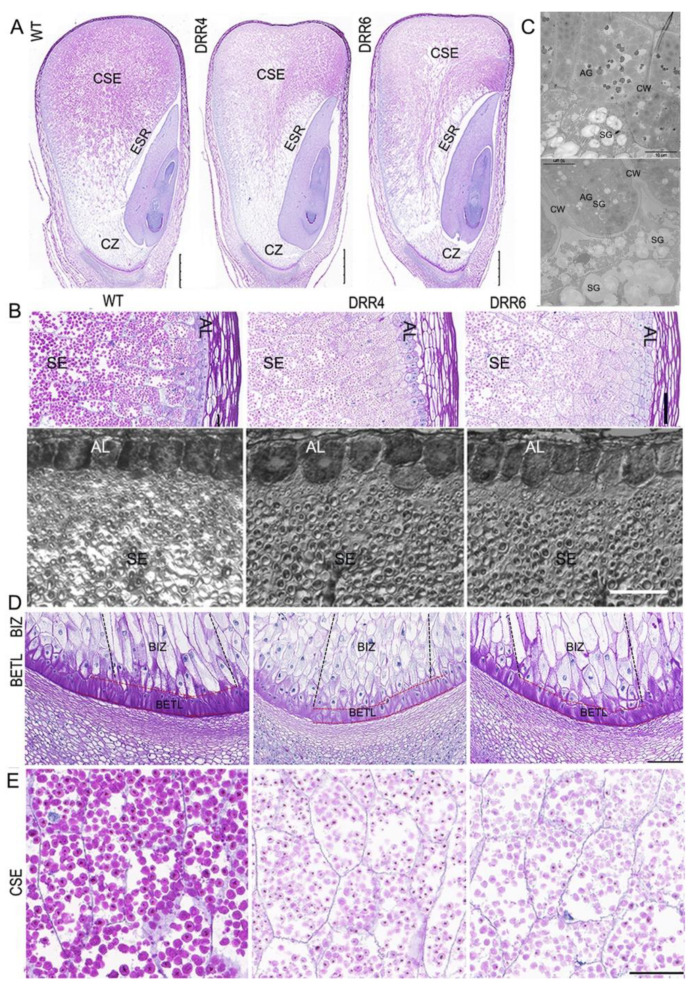
The cytological characterization of the developing maize kernel. (**A**): Comparison of the developing WT and the transgenic *ZmDRR206* overexpressor (*DRR-OE*) kernels at 18 DAP by paraffin sectioning. The longitudinal sections at the middle part of the 18 DAP kernels were stained by periodic acid-schiff stain (PAS). CSE, the central starch endosperm; ESR, embryo surrounding region; CZ, the conducting zone. Bar = 1 mm. (**B**): Comparison of the developing aleurone layer (AL) and starch endosperm (SE) cells in the developing WT and *DRR-OE* kernels at 18 DAP. Upper: the PAS-stained section, bar = 100 μm; lower: the semi-thin section, bar = 50 μm. (**C**): The cellular characterization of the AL cells under TEM. AG, aleurone grain; SG, starch granule; CW, cell wall. Bar = 10 μm. (**D**): Comparison of the developing basal endosperm transfer layer (BETL) and the basal intermediate zone (BIZ) in *DRR-OE* and WT kernels at 18 DAP. The length of both the BETL (between the red dashed lines) and BIZ (between the dark dashed lines) cells were shorter and the wall ingrowths of flange BETL cells were much less in *DRR-OE* kernels than that in WT kernel. (**E**): The PAS-stained granules appeared paler in color in the CSE cells of *DRR-OE* kernels than that in WT kernels. WT is the wild-type inbred line LH244, *DRR4* and *DRR6* are *ZmDRR206* overexpressing transgenic everts *DRR-OE4* and *DRR-OE6*. Bar = 100 μm in (**D**,**E**).

**Figure 5 ijms-24-08735-f005:**
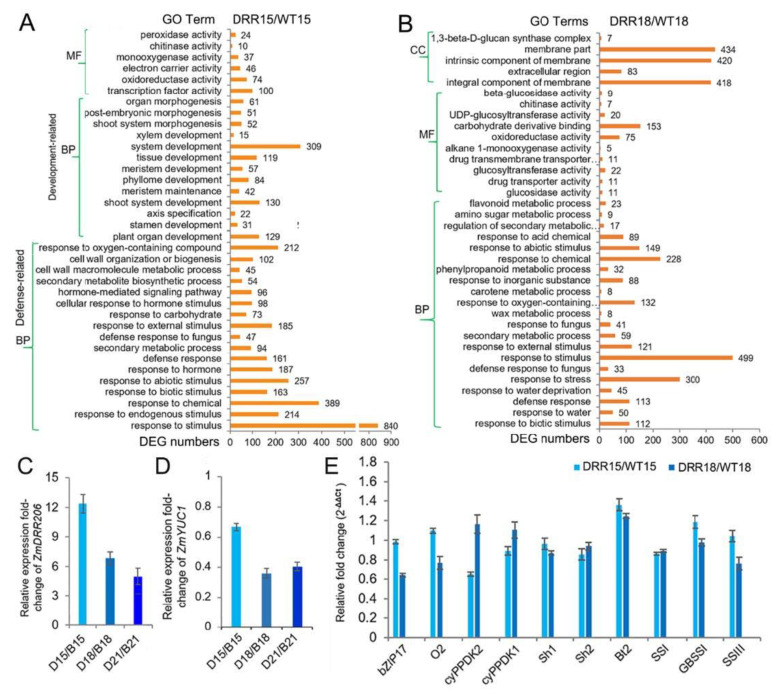
The transcriptome reprogramming induced by *ZmDRR206* overexpression in developing *DRR-OE* kernels. (**A**,**B**): The GO enrichment analysis of the DEGs between *DRR-OE* and WT kernels at 15 DAP (**A**) and at 18 DAP (**B**). CC, cellular component; MF, molecular function; BP, biological process. The numbers beside the column of the diagram are the numbers of the DEGs enriched in the functional categories. WT15/WT18 is the kernel from the wild-type inbred line LH244 at 15 DAP/18 DAP, DRR15/DRR18 is the kernel from the *ZmDRR206* overexpressors at 15 DAP/18 DAP. (**C**,**D**): The fold-change of the relative expression level of *ZmDRR206* (**C**) or *ZmYUC1* (**D**) between *DRR-OE* and WT kernels at 15, 18 and 21 DAP, respectively. B15/B18/B21 is the wild-type LH244 kernel at 15 /18 /21 DAP, respectively; D15/D18/D21 is the *DRR-OE* kernel at 15 /18 /21 DAP, respectively. (**E**): The fold-change of the relative expression level of kernel development or starch biosynthesis-related genes in the developing kernels of *DRR-OE* relative to WT by RT-qPCR. cyPPDK1/2, chloroplast pyruvate phosphate dikinase1/2; Sh1/2, shrunken1/2; Bt2, brittle-2; SSI, starch synthase I; GBSS1 (Waxy1), granule-bound starch synthase1; SSIII, starch synthase III. The rations of the expression level in DRR15/WT15 and DRR18/WT18 are detected in kernels from *DRR-OE* and WT at 15 DAP and 18 DAP, respectively. The statistical analysis in (**C**–**E**) was conducted using a paired Student’s *t*-test between WT and DRR at the indicated time point—15 DAP, 18 DAP or 21 DAP—in three replicates per genotype. DRR is DRR4 transgenic line. The significant differences are the absolute fold-change >2 under *p* < 0.05. That is the difference in expression level between WT and DRR of *ZmDRR206* and *ZmYUC1* was significant except *ZmYUC1* at 15 DAP, while the transcriptional difference of other genes was not significant.

**Figure 6 ijms-24-08735-f006:**
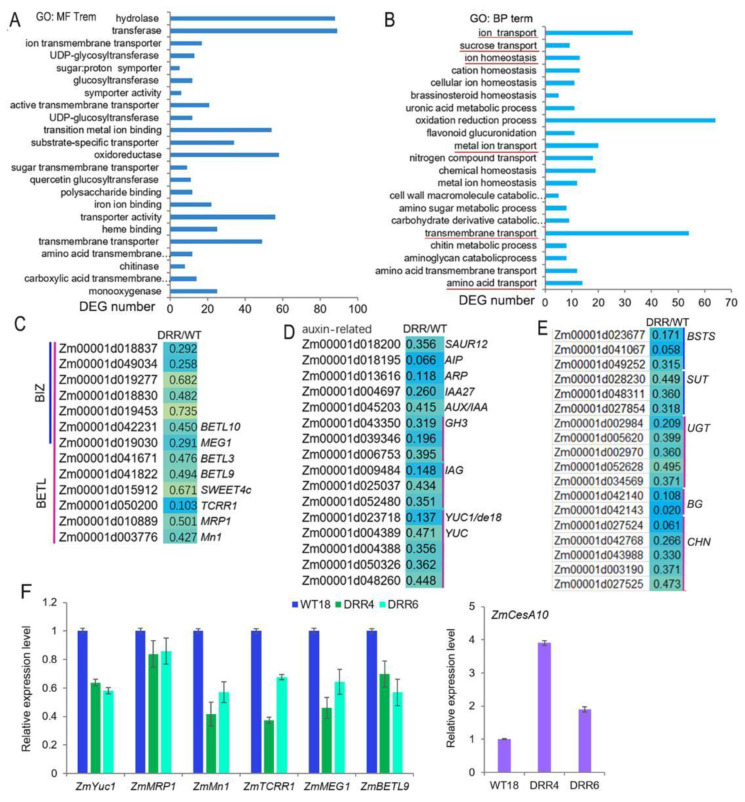
The expression of BETL development- and auxin signal-related genes was down-regulated by *ZmDRR206* overexpression in developing *DRR-OE* kernels. (**A**,**B**): The most significant enriched molecular function (**A**) and biological process (**B**) GO terms of the down-regulated DEGs based on RNA-seq analysis of the BETL tissue of *DRR-OE* and WT kernel at 18 DAP. The most important functional categories were underlined in red. (**C**–**E**): Heatmap showing changes in the expression levels of BETL development-related (**C**), auxin signal-related (**D**) and cell transportation and wall modification-related (**E**) genes in BETL tissue of *DRR-OE* and WT kernel. AIP, auxin-induced protein; ARP, auxin responsive protein; GH3, indole-3-acetic acid-amido synthetase; IAG, indole-3-acetate beta-glucosyltransferase; YUC, flavin monooxygenase; BETL, basal endosperm transfer layer; BIZ, basal intermediate zone. BSTS, bidirectional sugar transporter sweet13/14; SUT, sucrose transporter; UGT, UDP-glycosyltransferase; BG, beta-glucanase; CHN, chitinase. WT is the BETL tissue from wild-type LH244 kernel, DRR is the BETL tissue from *DRR-OE* kernel. The numerical values are the ratios of the DEGs in the comparison (DRR/WT), the background color showed the relative gene expression levels in the comparison (red represents up-regulated, blue is down-regulated, and yellow is no significant change). (**F**): The relative expression level of BETL-development-related genes in developing BETL of *DRR-OE* relative to that of WT kernel by RT-qPCR. WT18 is the BETL from the kernel of wild-type LH244 at 18 DAP, *DRR4* and *DRR6* are the BETL from the *DRR-OE* kernel at 18 DAP of the *ZmDRR206* overexpressing transgenic everts *DRR-OE4* and *DRR-OE6*. The statistical analysis was conducted using a paired Student’s *t*-test (between WT and DRR4 or DRR6) in three replicates per genotype. The significant differences are the absolute fold-change >2 under *p* < 0.05. That is, the transcriptional difference of most genes was not significant, except the transcriptional difference of *ZmMn1, ZmTCRR1* and *ZmCesA10* in WT and DRR4 was significant.

**Figure 7 ijms-24-08735-f007:**
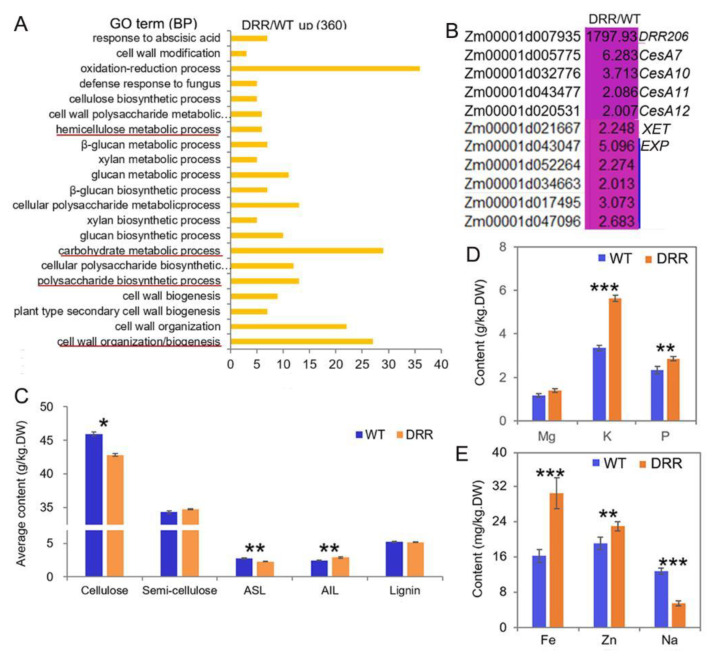
The cell wall organization/biogenesis-related functions were up-regulated in developing BETL of *DRR-OE* kernel. (**A**): The most significantly enriched biological process of the up-regulated DEGs from RNA-seq analysis of the BETL tissue of *DRR-OE* and WT kernel at 18 DAP. The most important functional categories were underlined in red. (**B**): Heatmap showing the fold-changes in expression levels of the cell wall biogenesis-related genes in BETL of *DRR-OE* relative to WT kernel at 18 DAP. *CesAs* were genes encoding cellulose synthase, *XET* encodes xyloglucan endotransglucosylase hydrolase; *EXP*s encode expansins. WT is the BETL from wild-type LH244 kernel, DRR is the BETL from *DRR-OE* kernel. The numerical values are the ratios of the DEGs in the comparison (DRR/WT), the background color showed the relative gene expression levels in the comparison (red represents up-regulated, blue is down-regulated, and yellow is no significant change). (**C**): The contents of the main cell wall components in the developing maize kernel at 18 DAP. ASL, acid soluble lignin; AIL, acid insoluble lignin; lignin is the summary of ASL and AIL. (**D**,**E**): The contents of the mineral elements in the air-dried mature maize kernels measured according to equal biomass. DW is dry weight. Data are presented as the mean values ± SD of three replicates per genotype. The asterisks represent a significant difference at * *p* < 0.05, ** *p* < 0.01, and *** *p* <0.001, according to a paired Student’s *t*-test.

## Data Availability

“MDPI Research Data Policies” at https://www.mdpi.com/ethics.

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
