# Peer review of "ZmDRR206 Regulates Nutrient Accumulation in Endosperm through Its Role in Cell Wall Biogenesis during Maize Kernel Development"

_ijms, 2023, doi:10.3390/ijms24108735_

Round 1

Reviewer 1 Report

Manuscript "ZmDRR206 regulates nutrient accumulation in endosperm through its role in cell wall biogenesis during maize kernel development" is very interesting.

General comments:
Authors showed that natural variations of ZmDRR206 were significantly associated with hundred-kernel weight (HKW) by association analysis of candidate genes, and ZmDRR206 overexpression in maize significantly decreased kernel size and HKW.

Detailed comments:
Quality of Figure 1 is very poor. Needs correction.
Figure 1: "p" not in superscript.
Figure 1: "r2" - what is it?
Figure 2D, E, F, H: The comparison with WT is not sufficient. A comparison should be made between all levels of the factor.
Figure 3C: The comparison with WT is not sufficient. A comparison should be made between all levels of the factor.
Figure 3D needs regression models.
Figure 5C, D, E need comparison and statistiacal testing.
Figure 6F needs comparison and statistiacal testing.

My suggestion:
"4.9. Statistical analysis": The comparison with WT is not sufficient. A comparison should be made between all levels of the factor.

Paper needs major revision.

Author Response

We greatly appreciate your efforts to handle our manuscript and to forward us the invaluable comments/suggestions of the reviewers. We provide our responses to all questions/suggestions one by one and implement all changes in the revised manuscript. We also checked the entire manuscript and made other modifications in the revised manuscript. Hopefully, you and the reviewers will be satisfied with these modifications and responses. We have uploaded the revised manuscript with tracked changes and the clean copy on the journal website in the online submission system. Any further comments will be greatly appreciated.

Sincerely Yours!

Jianrong Ye and co-authors

Revision comments

Manuscript "ZmDRR206 regulates nutrient accumulation in endosperm through its role in cell wall biogenesis during maize kernel development" is very interesting.
General comments:
Authors showed that natural variations of ZmDRR206 were significantly associated with hundred-kernel weight (HKW) by association analysis of candidate genes, and ZmDRR206 overexpression in maize significantly decreased kernel size and HKW.

Response: We greatly appreciate your comments on our manuscript and would like to take this opportunity to thank you for the constructive suggestions on our manuscript.

Detailed comments:
1. Quality of Figure 1 is very poor. Needs correction.
   Figure 1: "p" not in superscript.  Figure 1: "r2" - what is it?

Response: Thank you for the comment. We had revised Figure 1 with improved quality and changed “p” to “P” and r2 is linkage disequilibrium.

  1. Figure 2D, E, F, H: The comparison with WT is not sufficient. A comparison should be made between all levels of the factor.

Response: Thank you for the comment. We had revised this in Figure 2D, E, F, H by a one-way ANOVA followed by the least significance difference (LSD) test (P=0.05).

  1. Figure 3C: The comparison with WT is not sufficient. A comparison should be made between all levels of the factor.
    Response: Thank you for the comment. We had revised this in Figure 3C by a one-way ANOVA followed by the least significance difference (LSD) test (P=0.05).

  2. Figure 3D needs regression models.
    Response: Thank you for the comment. We had revised this in Figure 3D with regression models.

  3. Figure 5C, D, E need comparison and statistical testing. Figure 6F needs comparison and statistical testing.
    Response: Thank you for the comment. The statistical analysis in Figure 5C, D, E was conducted using a paired Student’s t-test between WT and DRR at indicate time-point (15-DAP, 18-DAP or 21-DAP)in three replicates per genotype. The significant differences are the absolute fold-change > 2 under P < 0.05. That is the difference in expression level between WT and DRR of ZmDRR206 and ZmYUC1 was significant except ZmYUC1 at 15-DAP, while the transcriptional difference of other genes was not significant. The statistical analysis in Figure 6F was conducted using a paired Student’s t-test (between WT and DRR4 or DRR6) in three replicates per genotype. The significant differences are the absolute fold-change > 2 under P < 0.05. That is, the transcriptional difference of most genes was not significant, except the transcriptional difference of ZmMn1, ZmTCRR1 and ZmCesA10 in WT and DRR4 was significant.

  1. My suggestion:"4.9. Statistical analysis": The comparison with WT is not sufficient. A comparison should be made between all levels of the factor.

Response: Thank you for the suggestions. We had revised this in all related place and in 4.9 with “All data with multiple levels were tested by multiple comparisons using the least significance difference (LSD) test.”.

Reviewer 2 Report

The study was focused on ZmDRR206 that regulates nutrient accumulation in endosperm through its role in cell wall biogenesis during maize kernel development. ZmDRR206 is a maize DIR that plays a role in maintaining cell wall integrity in the seedling growth and defense response in maize. Authors revealed that ZmDRR206 overexpression induced dysfunctional basal endosperm transfer layer (BETL) cells, which were shorter with less wall-ingrowth. Defense response was constitutively activated in developing maize kernel at 15- and 18-DAP by ZmDRR206 overexpression.

In my opinion, the work is quite interesting and important. In order to increase its scientific soundness, I recommend some improvements:

-        Introduction is very long, it should be shortened significantly,

-        Discussion part should be profoundly re-written and extended, with including some new citations in the research field,

-        Statistical analysis needs to be revised, factorial ANOVA with subsequent Tukey’s test should be used (because more than one factor was investigated),

-        Extensive editing of English language are required.

Extensive editing of English language are required. I suggest revision of the English style and grammar in the manuscript by the native speaker.

Author Response

We greatly appreciate your efforts to handle our manuscript and to forward us the invaluable comments/suggestions of the reviewers. We provide our responses to all questions/suggestions one by one and implement all changes in the revised manuscript. We also checked the entire manuscript and made other modifications in the revised manuscript. Hopefully, you and the reviewers will be satisfied with these modifications and responses. We have uploaded the revised manuscript with tracked changes and the clean copy on the journal website in the online submission system. Any further comments will be greatly appreciated.

Sincerely Yours!

Jianrong Ye and co-authors

The study was focused on ZmDRR206 that regulates nutrient accumulation in endosperm through its role in cell wall biogenesis during maize kernel development. ZmDRR206 is a maize DIR that plays a role in maintaining cell wall integrity in the seedling growth and defense response in maize. Authors revealed that ZmDRR206 overexpression induced dysfunctional basal endosperm transfer layer (BETL) cells, which were shorter with less wall-ingrowth. Defense response was constitutively activated in developing maize kernel at 15- and 18-DAP by ZmDRR206 overexpression.

In my opinion, the work is quite interesting and important. In order to increase its scientific soundness, I recommend some improvements:

Response: We greatly appreciate your comments on our manuscript and would like to take this opportunity to thank you for the constructive suggestions on our manuscript.

-        Introduction is very long, it should be shortened significantly,

   Response: Thank you for the comment. We had revised the Introduction and shortened a few paragraphs.

-        Discussion part should be profoundly re-written and extended, with including some new citations in the research field,

   Response: Thank you for the suggestions. We would pay more attention to this part and tried to re-write it and extend it with new citations.

-        Statistical analysis needs to be revised, factorial ANOVA with subsequent Tukey’s test should be used (because more than one factor was investigated),

   Response: Thank you for the suggestions. We had revised the statistical analysis with “All data with multiple levels were tested by multiple comparisons using the least significance difference (LSD) test.”.

-        Extensive editing of English language are required.

   Response: Thank you for the suggestions. The manuscript would be sent out for English language editing according to the MDPI webside.

Dear Reviewer: I am sorry for this response in a great hurry. As I have to go out for a week for the wedding of my niece, and the journal gave us 10days to revise the ms, the other members had also gone out. We will take careful to revise the ms in a week later according to your good suggestions.

thank you very much!

Your sincerely, Ye

Reviewer 3 Report

Maize is one of the most important agricultural crops, so the high interest in studying this crop is understandable. The authors of the submitted manuscript conducted a study with the little-known ZmDRR206 gene. The experiment was interesting, the results are reliable, the conclusions correspond to the results obtained. However. there are comments and recommendations.

1. The introduction is very detailed, I recommend shortening it

2. Specify that you used 6 transgenic maize lines and 6 of their T4 homozygous progenies. From which DRR-OE line?

3. Figure 2. Why use the amount of aa in DW as g/kg, usually µg/mg. Specify which DRR line?

4. Specify the reference to Arabidopsis (after Figure 2I)

5. Specify the reference for the effect of xenia on the endosperm

6. What is the DRR line in Figure 5

7. In figure 5, indicate DRR-OE, as in the caption to the figure

8. The same remarks are in Figure 7.

9. No link to the text of the manuscript information about bHLH and its reference does not match the list

10. Check the order of the references.

11. References to issue in accordance with the requirements of the journal

12. Reference (Wang et al) to issue according to the requirements of the journal

13. In section Materials and Methods 4.5. The given references should be indicated after the description of the corresponding method.

14. No description of the definition of composition

Author Response

We greatly appreciate your efforts to handle our manuscript and to forward us the invaluable comments/suggestions of the reviewers. We provide our responses to all questions/suggestions one by one and implement all changes in the revised manuscript. We also checked the entire manuscript and made other modifications in the revised manuscript. Hopefully, you and the reviewers will be satisfied with these modifications and responses. We have uploaded the revised manuscript with tracked changes and the clean copy on the journal website in the online submission system. Any further comments will be greatly appreciated.

Sincerely Yours!

Jianrong Ye and co-authors

Maize is one of the most important agricultural crops, so the high interest in studying this crop is understandable. The authors of the submitted manuscript conducted a study with the little-known ZmDRR206 gene. The experiment was interesting, the results are reliable, the conclusions correspond to the results obtained. However. there are comments and recommendations.

Response: We greatly appreciate your comments on our manuscript and would like to take this opportunity to thank you for the constructive suggestions on our manuscript.

  1. The introduction is very detailed, I recommend shortening it

Response: Thank you for the suggestions. We had revised the Introduction and tried to shorten it appropriately.

  1. Specify that you used 6 transgenic maize lines and 6 of their T4 homozygous progenies. From which DRR-OE line?

Response: Thank you for the comment. We are sorry to make such a confusion, in fact, we obtained six independent transgenic maize events, and three of them were cultured and their T4 homozygous progenies were used for experiments and named as DRR-OE.

  1. Figure 2. Why use the amount of aa in DW as g/kg, usually µg/mg. Specify which DRR line?

Response: Thank you for the suggestions. The used g/kg is equal to the usually used µg/mg, we think the content of amino acid is in massive amount, not in trace amount, thus g/kg was used. And the three independent lines, DRR3, 4 and 6 were used for here replicates for amino acid measuring.

  1. Specify the reference to Arabidopsis (after Figure 2I)

Response: Thank you for the suggestions. We had added the reference.

  1. Specify the reference for the effect of xenia on the endosperm

Response: Thank you for the suggestions. We had added the reference. 

  1. What is the DRR line in Figure 5

Response: Thank you for the comment. We had tested this with two DRR lines and presented the results of DRR4.

  1. In figure 5, indicate DRR-OE, as in the caption to the figure

Response: Thank you for the suggestions. We had revised it.

  1. The same remarks are in Figure 7.

Response: Thank you for the suggestions. We had revised it.

  1. No link to the text of the manuscript information about bHLH and its reference does not match the list

Response: Thank you for the comment. The basic helix-loop-helix (bHLH) transcriptional factor (TF), O11, functions as a central regulator in the gene networks governing maize endosperm development, nutrient metabolism, and stress response. The related reference is [49].

  1. Check the order of the references.

Response: Thank you for the suggestions. We had checked and revised it.

  1. References to issue in accordance with the requirements of the journal

Response: Thank you for the suggestions. We had revised it.

  1. Reference (Wang et al) to issue according to the requirements of the journal

Response: Thank you for the suggestions. We had revised it to [32].

  1. In section Materials and Methods 4.5. The given references should be indicated after the description of the corresponding method.

Response: Thank you for the suggestions. We had revised it.

  1. No description of the definition of composition

Response: Thank you for the suggestions. We had checked the text and composition could be found, if it is the cell wall main components, it was cellulose, semi-cellulose, ASL, AIL and lignin (the total of ASL and AIL).

Reviewer 4 Report

Authors submitted a manuscript entitled “ZmDRR206 regulates nutrient accumulation in endosperm through its role in cell wall biogenesis during maize kernel development” to IJMS. The authors studied the role of ZmDRR206, a DIR protein in maize kernel development. They found that ZmDRR206 plays a dominant role in nutrient storage accumulation during kernel development, and overexpression of ZmDRR206 resulted in smaller and shrunken maize kernels with reduced starch content and HKW. They observed dysfunctional basal endosperm transfer layer cells and downregulation of BETL-development and auxin signal-related genes, while cell wall biogenesis-related genes were upregulated. These findings suggest that ZmDRR206 regulates cell development, nutrient metabolism, and stress responses during maize kernel development. The manuscript is well balanced and reports good results. I have a few comments and suggestions.

Section 2.2. The shrunken and collapsed …. Re-write this sentence and try not to repeat the same word more times.

In the last paragraph of the discussion section, add a few lines about the critical findings of this study, remaining research findings, and possible future plans that can be drawn from the current results. In addition, try to speculate the results and discuss them with relevance to the previously reported results. Authors have provided the discussion section almost like the results section. Discussion needs substantial improvements. Mainly the statements about the critical evaluation of current findings in the context of previous research and some unresolved questions or areas of uncertainty that may require further investigation are missing.

Please explain how the differential expression was performed using FPKM values? FPKM (Fragments Per Kilobase Million) values are not ideal for differential expression analysis, which compares gene expression between two or more conditions. This is because FPKM normalization assumes that the total amount of RNA in the sample is constant across samples, which may not be true if there are biological differences between the samples, such as different cell types or experimental treatments.

There are a few mistakes regarding sentences and grammar. Overall, the writing is fine.

Author Response

Comments and Suggestions for Authors:

Authors submitted a manuscript entitled “ZmDRR206 regulates nutrient accumulation in endosperm through its role in cell wall biogenesis during maize kernel development” to IJMS. The authors studied the role of ZmDRR206, a DIR protein in maize kernel development. They found that ZmDRR206 plays a dominant role in nutrient storage accumulation during kernel development, and overexpression of ZmDRR206 resulted in smaller and shrunken maize kernels with reduced starch content and HKW. They observed dysfunctional basal endosperm transfer layer cells and downregulation of BETL-development and auxin signal-related genes, while cell wall biogenesis-related genes were upregulated. These findings suggest that ZmDRR206 regulates cell development, nutrient metabolism, and stress responses during maize kernel development. The manuscript is well balanced and reports good results. I have a few comments and suggestions.

Response: We greatly appreciate your comments on our manuscript and would like to take this opportunity to thank you for the constructive suggestions on our manuscript.

Section 2.2. The shrunken and collapsed …. Re-write this sentence and try not to repeat the same word more times.   

Response: Thank you for the suggestions. This sentence was revised to “The aberrant endosperm with substantially reduced starch content is usually associated with reduced number of starch grain (SG) or smaller SG in diameter.” In this version of text.

In the last paragraph of the discussion section, add a few lines about the critical findings of this study, remaining research findings, and possible future plans that can be drawn from the current results. In addition, try to speculate the results and discuss them with relevance to the previously reported results. Authors have provided the discussion section almost like the results section. Discussion needs substantial improvements. Mainly the statements about the critical evaluation of current findings in the context of previous research and some unresolved questions or areas of uncertainty that may require further investigation are missing.   

Response: Thank you for the suggestions. We had ever tried to explain the related results in the first part 3.1 of Discussion, and speculate the results and discuss them with relevance to the previously reported results in the second part 3.2 of Discussion. As for the critical findings of this study and possible future plans, we had added in 5. Conclusion section. We had tried our best to improve Discussion section and had made much revision in the statements about the critical evaluation of current findings in the context of previous research. Hopefully, you will be satisfied with these modifications and responses. As for the remaining research findings is the result about the effect of ZmDRR206-overexpression on maize ear rot by artificial inoculation was not significant as that of stalk rot and it is not easy to interpret. Indeed, we are preparing to measure sugar and auxin content in the BETLs and endosperm of the developing kernel, that is what may require further investigation.

Please explain how the differential expression was performed using FPKM values? FPKM (Fragments Per Kilobase Million) values are not ideal for differential expression analysis, which compares gene expression between two or more conditions. This is because FPKM normalization assumes that the total amount of RNA in the sample is constant across samples, which may not be true if there are biological differences between the samples, such as different cell types or experimental treatments.

Response: Thank you for the comment. We had checked the original RNA-Seq data and found the differential expression analysis was performed with DESeq2, which used corrected readcount (Calculated based on the shrinkage model of the difference analysis software) to obtain log2FoldChange. FPKM is usually used to compare the relative expression level of different genes in the same sample. Thus, the differential expression analysis of some important genes was further confirmed by RT-qPCR analysis.

Reviewer 5 Report

The manuscript addresses an important topic regarding the role of the maize dirigent protein namely ZmDRR206 in kernel development and endosperm nutrient content. The study falls within the scope of the special issue providing new insights on maize kernel development mechanisms and strategies to improve crop yield.  The experimental design is well organized and presented and the manuscript is well written highlighting an enormous amount of laborious research utilizing current molecular methods and tools. However, there are some minor issues that should be elaborated to improve the manuscript so that the reader could easily appreciate the wealth of information and the impact of the study. These are the following:

1.      In the Introduction section on the last paragraph the statement “in our previous research” should be complemented by relevant citations.

2.      In the same paragraph citation [40] should be completed in the references list with Journal issue, pg and DOI number.  As the term (under review) is not appropriate for published articles.

3.      In Results section 2.1 it would be nice to show the polymorphism of the single SNP within the coding region (chr2.S_243395225, v4) that is associated with HKW.

4.      In Fig 2 (A) it would be nice to indicate with thin arrows the shrunken kernels of DRR3, DRR4 and DRR6 ears. Similarly in Fig.4 (D) thin arrows could indicate the differences in BIZ and BETL cells in DRR4 and DRR6 endosperm tissues.

5.      The Discussion section title should be in singular.  At the end of the discussion, it would be nice to add a closing statement regarding the impact of ZmDRR206 role in maize kernel size and crop yield and how further research could improve maize production under the climate changing environment. Alternatively, this statement could be placed at the end of the Conclusion section.

6.      In the Conclusion section:

a.      The first line states that “ZmDRR206, in regulating maize kernel development” should be rephrased to “ZmDRR206, in kernel development regulation”.

b.      The statement “The natural variations….” should be rephrased to “Gene association analysis of maize germplasm revealed that significant SNPs associated with HKW were detected in the ZmDRR206 gene region”.

c.       “and defense response” should be followed by a stop and short sentences as follows : “Its overexpression…BETL-development”. “Also, auxin signal-related constitutively up-regulated defense responses…. and significantly reduced contents of cellulose and acid soluble lignin (the main lignin in cell wall) in developing maize kernels”.  “These transcriptional …ultimately led the mature DRR-OE maize kernel to be smaller, opaque and shrunken… and HKW compared to the wild type”.

7.      Check the references as all should be written in the same uniform style (i.e., terms in bold font). Similar bold font is shown in the Supplemental Figures legends in the main manuscript.

8.      In Supplementary Table 4.  The direction of the primers’ sequences should be indicated (i.e., 5’-3’). Also, it is nice to show all sequences in capital letters. The title of column D should be shown.

An exceptional manuscript.

Author Response

The manuscript addresses an important topic regarding the role of the maize dirigent protein namely ZmDRR206 in kernel development and endosperm nutrient content. The study falls within the scope of the special issue providing new insights on maize kernel development mechanisms and strategies to improve crop yield.  The experimental design is well organized and presented and the manuscript is well written highlighting an enormous amount of laborious research utilizing current molecular methods and tools. However, there are some minor issues that should be elaborated to improve the manuscript so that the reader could easily appreciate the wealth of information and the impact of the study. These are the following:

Response: We greatly appreciate your positive comments on our manuscript and would like to take this opportunity to thank you for the constructive suggestions on our manuscript.

  1. In the Introduction section on the last paragraph the statement “in our previous research” should be complemented by relevant citations.

Response: Thank you for the suggestions. It is very unhappy to remind this manuscript of our previous work, as it had run a long long way under reviewing for more than one year, and it is now can be viewed in Research Square Platform LLC, 2023. We are sorry for this situation, and I hope it will end in next month for journal of Communications Biology as three months has passed and two rounds of review has finished.

  1. In the same paragraph citation [40] should be completed in the references list with Journal issue, pg and DOI number.  As the term (under review) is not appropriate for published articles.

Response: Thank you for the suggestions. We are sorry for such a problem and I hope it will end in next month.

  1. In Results section 2.1 it would be nice to show the polymorphism of the single SNP within the coding region (chr2.S_243395225, v4) that is associated with HKW.

Response: Thank you for the suggestions. The single SNP within the coding region was added in Figure 1.

  1. In Fig 2 (A) it would be nice to indicate with thin arrows the shrunken kernels of DRR3, DRR4 and DRR6 ears. Similarly in Fig.4 (D) thin arrows could indicate the differences in BIZ and BETL cells in DRR4 and DRR6 endosperm tissues.

Response: Thank you for the suggestions. Thin arrows were added to indicate the shrunken kernels of DRR3, DRR4 and DRR6 ears in Figure 2A, and thin arrows were added in Figure 4D to indicate the differences in BIZ and BETL cells in DRR4 and DRR6 endosperm tissues.

  1. The Discussion section title should be in singular.  At the end of the discussion, it would be nice to add a closing statement regarding the impact of ZmDRR206 role in maize kernel size and crop yield and how further research could improve maize production under the climate changing environment. Alternatively, this statement could be placed at the end of the Conclusion section.

Response: Thank you for the suggestions. The Discussion section title 3.1 and 3.2 was in singular. And a statement “Although ZmDRR206 overexpression increased disease resistance, enhanced drought tolerance in maize, it also resulted in small kernel and diminished seedling growth, the same negative effects as other disease resistance genes. To alleviate or eliminate the growth suppression side-effect of ZmDRR206-overexpression on maize seedling growth and kernel development, it would be beneficial to put ZmDRR206 under the control of an inducible promoter to make it possible that the expression of ZmDRR206 would be induced to increase rapidly upon pathogen infection or drought/heat stress, while remain basal expression under normal growth condition for adaptation to the climate changing environment. ” was added at the end of the conclusion section.

  1. In the Conclusion section:
  2. The first line states that “ZmDRR206, in regulating maize kernel development” should be rephrased to “ZmDRR206, in kernel development regulation”.

Response: Thank you for the suggestions. We appreciated your careful and friendly suggestions very much! This was revised in this version of text.

The statement “The natural variations….” should be rephrased to “Gene association analysis of maize germplasm revealed that significant SNPs associated with HKW were detected in the ZmDRR206 gene region”.

Response: Thank you for the suggestions. We appreciated your careful and friendly suggestions very much! This was revised in this version of text.

“and defense response” should be followed by a stop and short sentences as follows : “Its overexpression…BETL-development”. “Also, auxin signal-related constitutively up-regulated defense responses…. and significantly reduced contents of cellulose and acid soluble lignin (the main lignin in cell wall) in developing maize kernels”.  “These transcriptional …ultimately led the mature DRR-OE maize kernel to be smaller, opaque and shrunken… and HKW compared to the wild type”.

Response: Thank you for the suggestions. We appreciated your careful and friendly suggestions very much! These was revised in this version of text.

  1. Check the references as all should be written in the same uniform style (i.e., terms in bold font). Similar bold font is shown in the Supplemental Figures legends in the main manuscript.

Response: Thank you for the suggestions. These was revised in this version of text.

  1. In Supplementary Table 4.  The direction of the primers’ sequences should be indicated (i.e., 5’-3’). Also, it is nice to show all sequences in capital letters. The title of column D should be shown.

Response: Thank you for the suggestions. The direction of the primers’ sequences had been indicated and all sequences were changed into capital letters.

Round 2

Reviewer 1 Report

Thanks to the authors for their efforts to improve the manuscript. All is ok.

Author Response

Dear reviewers,

We greatly appreciate your efforts to handle our manuscript and to forward us the invaluable comments/suggestions of the reviewers.  Any further comments will be greatly appreciated.

Sincerely Yours!

Jianrong Ye and co-authors

Reviewer 4 Report

The authors addressed all the comments. Now the manuscript is more balanced and appropriate for further consideration. 

Overall Quality of the English language is fine but minor editing is still required.